# Vaccines for Non-Viral Cancer Prevention

**DOI:** 10.3390/ijms222010900

**Published:** 2021-10-09

**Authors:** Cristina Bayó, Gerhard Jung, Marta Español-Rego, Francesc Balaguer, Daniel Benitez-Ribas

**Affiliations:** 1Department of Immunology, Fundació Clínic per la Recerca Biomèdica (FCRB), Hospital Clínic Barcelona, University of Barcelona, 08036 Barcelona, Spain; bayo@clinic.cat (C.B.); espanol@clinic.cat (M.E.-R.); 2Centro de Investigación Biomédica en Red en Enfermedades Hepáticas y Digestivas (CIBEREHD), Department of Gastroenterology, IDIBAPS (Institut d’Investigacions Biomèdiques August Pi i Sunyer), Hospital Clínic Barcelona, University of Barcelona, 08036 Barcelona, Spain; jung@clinic.cat

**Keywords:** vaccines, dendritic cells, cancer, prevention, Lynch syndrome

## Abstract

Cancer vaccines are a type of immune therapy that seeks to modulate the host’s immune system to induce durable and protective immune responses against cancer-related antigens. The little clinical success of therapeutic cancer vaccines is generally attributed to the immunosuppressive tumor microenvironment at late-stage diseases. The administration of cancer-preventive vaccination at early stages, such as pre-malignant lesions or even in healthy individuals at high cancer risk could increase clinical efficacy by potentiating immune surveillance and pre-existing specific immune responses, thus eliminating de novo appearing lesions or maintaining equilibrium. Indeed, research focus has begun to shift to these approaches and some of them are yielding encouraging outcomes.

## 1. Introduction

The idea of cancer vaccines is not new. In fact, it has been studied intermittently over the last four decades. The first antitumoral vaccine clinical trials focused on the identification and targeting of tumor-associated antigens (TAAs) in therapeutic settings, either with synthetic peptides [1,2,3] or with whole tumor cells or cell lysates [4]. Although these approaches elicited minimal impact on overall survival [5], efforts to develop such therapies have persevered over the years due to the high potential of immunotherapeutic vaccines to elicit and amplify antigen-specific immune responses.

With the rise of new insights into the tumor-suppressive microenvironment (TME), immunotherapeutic approaches targeting inhibitory molecules started to appear. Amongst them was the extraordinarily successful immune checkpoint therapy (ICP) with antibodies targeting cytotoxic T lymphocyte-associated antigen 4 (CTLA4) or programmed death 1/ligand 1(PD1)/(PDL1), which overcame immune suppression and induced sustained regression of disease in a subset of patients with cancer [6]. Even though still to date many cancer patients do not respond to them and elicit immune-related adverse events [7], new data from trials based on checkpoint inhibitors has led to the appearance of multiple combinatory therapeutic approaches with cancer vaccines and to a subsequent increased overall therapeutic efficacy in the last few years [8,9,10]. Although there are promising synergetic data from multiple trials (reviewed in [11]), the clinical effectiveness of cancer treatment vaccines still remains low [12]. 

This lack of success has been studied in various reviews based on a meta-analysis of accumulated data from a large number of conducted clinical trials [13,14,15]. Although there are many promising early trials and combination studies currently underway [15], there is a common denominator when addressing the major problems of therapeutic vaccines: the late stage of the disease at which treatment is administered. This difficulty can be solved with a preventive approach, where vaccination is performed prophylactically before cancer appearance or in early disease stages, as it has been proven that pre-neoplastic lesions can be controlled by an optimal immune response [16]. A healthy individual with an intact immune system and a high risk of cancer would make an ideal target in this context. 

Cancer preventive vaccines (CPV) are commonly associated with the prevention of oncogenic viruses. Indeed, the most successful CPV to date confer protection against neoplasms derived from infection of human papillomavirus (HPV) [17], which is associated with tumors such as cervical cancer, head and neck squamous cell carcinoma (HNSCC) and anal cancer; hepatitis B virus (HBV) [18], that has been a proven cause of hepatocellular carcinoma (HCC); and Epstein-Barr virus (EBV) [19], the first human tumor virus discovered and associated with epithelial-, lymphocyte-, and smooth muscle-derived tumors. However, the majority of tumors have unknown or nonexistent infectious etiology.

Up until recently, the idea of CPV against non-viral cancers has been associated with a high risk of autoimmunity because in the vast majority of clinical trials the targeted antigens were TAAs, which generally derive from self-antigens that are selectively or over-expressed in tumor cells or involved in tissue differentiation such as the transmembrane glycoprotein Mucin 1 (MUC1) [20]. Although several CPV trials focused on TAAs have shared promising data, the fear of breaking self-tolerance in healthy tissues or cells drove the focus to therapeutic settings, where the risks associated with these target molecules are more acceptable.

Due to the above, CPVs were at an impasse until not too long ago: with the rise of new technologies in high-throughput sequencing (such as next-generation sequencing, NGS) and the improvement of artificial intelligence (AI)-based biological prediction algorithms, in silico predicted and in vitro validated neoantigen-based vaccines have appeared and brought new hope in the landscape of preventive (and therapeutic) cancer vaccines and personalized medicine [21,22,23]. 

In this review, we will give an overview of the current state of CPVs, discuss their still barely tested potential in humans and highlight key biological and manufacturing points for vaccine improvement. We review the previous work on cancer therapies and how to implement them into cancer prevention, giving emphasis to the recently neoantigen-based vaccines and proposing Lynch Syndrome patients as an ideal clinical model to study these approaches.

## 2. Immunology of Cancer Prevention Vaccines

There are several points to consider regarding immunological responses to confer durable and optimal protection when designing a CPV. Ideally, a CPV search to modulate the host’s immune system to induce durable and protective immune memory against cancer-related antigens so that it can eliminate de novo appearing lesions. This is achieved by an optimal induction of innate immune responses, the uptake of antigens by antigen-presenting cells (APCs) such as dendritic cells (DCs), and the subsequent priming of T cells by MHC-TCR interactions in draining lymph nodes [24]. Correct priming is crucial for the induction of either de novo or pre-existing specific CD4+ helper and CD8+ cytotoxic T cell responses, which will eventually generate long-lived central memory T (TCM) cells and effector memory T (TEM) cells. Importantly, pre-existing specific immune responses introduce the concept of immune surveillance, which establishes that the immune system is constantly surveying the body for abnormal or transformed cells which are detected through specific antigenic molecules. During tumoral development, cells undergo a process called immunoediting, where due to the selective pressure of immunosurveillance, those cells with immunosuppressive and evasive characteristics that favor the tumor microenvironment (TME) (e.g., loss of MHC-I expression, PD-L1 overexpression, TGF-b release, etc.) are selected [12]. To avoid it, immunosurveillance could be reinforced with the administration of vaccines targeting these tumoral antigens. 

Amongst the many factors that can impact the above-mentioned response cascade, the human leukocyte antigen (HLA) haplotype is critical, especially in neoantigen-based vaccines. HLAs, the molecules responsible to present epitopes derived from extracellular and intracellular antigens to the immune system to activate specific CD8+ (HLA-I) and CD4+ (HLA-II) responses, have a characteristically hypervariable nature that impacts antigen-presenting efficiency depending on peptide sequence [24]. This introduces variability to the response against a determinate vaccine and a target population restriction when working with in silico-predicted peptide vaccines. For example, several recent neoantigen vaccine clinical trials require the HLA-A*02:01 allele, the most common HLA allele in the European population (Figure 1), as inclusion criteria [25,26,27] in order to ensure optimal induced immunogenicity. It is important to consider these molecules either when designing the vaccine, when performing validation in vitro assays, and within the individuals included in the trial to obtain robust data and to avoid false negatives regarding immunogenicity.

Secondly, antigen choice is also highly impactful in the resulting immune responses. Concerning this, neoantigens have been suggested to be more immunogenic than TAAs and demonstrated to increase lymphocytic infiltration, probably due to being specific to cancer cells [29]. Neoantigens are molecules derived from mutations occurring only in malignant or non-healthy cells. Due to their non-self-nature, they do not appear in healthy tissues and thus are also absent in the thymus, meaning they do not have a previously built immune central tolerance that most TAAs have. For cancer prevention, neoantigens present far less associated risks than TAAs as there are normally no related autoimmunity events—which makes neoantigen-based vaccines more feasible to being accepted into clinical trials. In peptide vaccines, epitope length also impacts the resulting immune response. Vaccines based on synthetic long peptides (SLPs), which range from 20–35 amino acids in size and are derived from two or more of the patient’s tumor-specific mutant antigens, seem to exert mainly CD4+ cellular responses [26]. This is probably due to the compatibility of long peptides to being processed as extracellular antigens and thus presented via MHC-II molecules to CD4+ lymphocytes. As an advantage, SLPs are generally not restricted to bind specific HLAs thus avoiding target population limitations. On the contrary, vaccines based on epitopes restricted in length to either MHC-I (8 to 9 amino acids) or MHC-II molecules (12 amino acids or more) generally induce both CD4+ and CD8+ responses, although their effectiveness is normally restricted to determinate MHC alleles. The open-ended MHC-II binding groove may be related to this fact, as the low restrictive nature of the molecule allows the presentation of a greater variety of peptides [30]. The recent reports on the non-overlapping role of neoantigen responses mediated by CD4+ and CD8+ T cells [31] give insight on how much impact the antigen type and length choice have in eliciting a complete antitumoral response by CPVs. 

Differences in the stimulation and antigen processing in APCs, especially DCs, are also known to heavily influence T-cell responses [32]. As key regulators of the immune response, DCs play a central role in antitumoral immunity by releasing immunomodulatory cytokines and attracting and priming specific T-cells [33]. Moreover, their well-known ability for cross-presentation (in sum, being able to present extracellular endocytosed antigens through the MHC-I pathway) renders them paramount for the induction of CD8+ Th1 responses [34], which should be considered when designing antitumoral CPVs. Along the same lines, repetitive antigen stimulation has been demonstrated to affect the expression and functionality of memory T cells [35]. Particularly in prophylactic approaches (although also importantly in therapeutic vaccines), the timing of vaccination should be considered carefully as it has the potential to change the resulting T-cell phenotypes [23].

Finally, it is important to address the question of how vaccine efficacy and impact on disease prevention will be evaluated in clinical trials. Due to the nature of this approach, there are no measurable outcomes except tumor development. Patient follow-up timings substantially differ among different studies, normally ranging from 2 to 5 years, even 10 in some cases. Therefore, apart from the differences in immunological responses between the study arms and pre- and post-vaccination, further consideration should be given to outcome measuring. For instance, standardization of a follow-up time period or researching more advanced clinical models that could be helpful in determining vaccine efficacy prior to vaccination in humans. Along these lines, organoid models, which are tissue-derived adult stem cells embedded into a 3D matrix and grown with high efficiencies into self-organizing organotypic structures [36], have been demonstrated to recreate the architecture and physiology of human organs and diseases in very close detail [37,38], which could be useful in vaccine efficacy determination.

## 3. Antigen Targets

Preventive approaches can be divided into various levels. In general, primary prevention is focused on avoiding initiation of disease, which would include either prevention in healthy patients and the targeting of pre-malignant lesions, while secondary prevention is directed to prevent relapses of secondary primary cancers, such as in contralateral breast cancer, or to detect and treat early non-invasive primary tumors. Lastly, tertiary prevention would include treating a post-therapy patient to prevent recurrence of that disease [39]. There are several clinical trials for CPVs covering all prevention levels, and target antigens vary depending on which stage the vaccine is acting on, summed up in Table 1. 

### 3.1. Tumor Associated Antigens (TAAs)

TAAs are a type of antigens that derive from self-antigens but are selectively or over-expressed in tumor cells or involved in tissue differentiation such as MUC1 [20]. This category also includes cancer testis antigens (CTAs), which are normally expressed in reproductive tissues (fetal, ovaries, testes, trophoblasts) and have a restricted presence on other healthy tissues. These antigens are ectopically expressed in several types of tumors and as germ cells are devoid of HLA-I molecules, they are recognized as exogen by the immune system [40,41]. 

As mentioned before, the use of these type of molecules have been historically associated to an elevated risk of autoimmunity, but many of them have already been successfully implemented in cancer therapies with minimal adverse effects, such as the molecules MUC1 for epithelial tumors [42] or the human epidermal growth factor receptor 2 (HER2) in breast cancer [43] among many others. In a primary prophylactic setting, excepting individuals at high cancer risk, these antigens could be problematic due to the absence of a developed lesion where the molecules are aberrantly expressed, so inducing immunogenicity against them would mean targeting mainly healthy tissue leading to a high theoretical risk of autoimmunity. Moreover, the expression of TAAs on healthy cells can also lead to the erasure of high-affinity T cells in the thymus during ontogeny, which may compromise vaccine effectiveness [21,44]. However, in pre-neoplastic lesions or in a secondary preventive level where patients are at a high risk of disease relapse or recurrence these strategies could be more beneficial.

MUC1, a molecule aberrantly glycosylated and overexpressed in a variety of epithelial cancers [20], was one of the first TAAs to be tested in a clinical trial of therapeutic cancer vaccines in patients with metastatic colon, pancreatic, and breast cancer [1]. In another setting and in murine models, Olivera J. Finn et al. showed that vaccination with MUC1 delays inflammatory bowel disease and prevents progression to colitis-associated colon cancer [45], which drew attention to preventive approaches instead of therapeutical ones. Of the 324 trials with the synthetic MUC1 peptide registered in Clinicaltrials.gov, even though the molecule has induced different levels of specific and durable immune responses, there are still no successful clinical results as a therapeutic option in humans [42]. The lack of clinical success has been attributed to the tumor immunosuppressive microenvironment in advanced cancer patients. However, there are several trials with promising preventive approaches: in a phase II trial completed in 2012 (NCT00773097) in individuals with a previous history of advanced adenomatous polyps (the most frequent premalignant lesion in the colon), vaccination with the drug product MUC1 peptide-poly-ICLC vaccine (containing the MUC1 peptide and the adjuvant poly-ICLC) elicited MUC1-specific immune responses and immune memory that were not associated with any toxicity. No clinical endpoints were assessed [46]. Importantly, unresponsive individuals were found to have elevated numbers of myeloid-derived suppressor cells in pre-vaccination PBMCs (peripheral blood mononuclear cells), indicating that active immunosuppression was already present in pre-malignant lesions and thus suggesting a greater benefit for this vaccine in earlier disease or in healthy individuals with a known risk for colorectal cancer [46]. Also using the same drug, a recent randomized phase II clinical trial is studying its efficacy on patients with newly diagnosed advanced adenomatous colorectal polyps for the prevention of CRC and polyp recurrence (NCT02134925) with no relevant clinical results available yet. Lastly, in 2017, a phase I trial with the MUC1 peptide-Poly-ICLC vaccine from the National Cancer Institute (NCT03300817) was initiated with the objective of preventing lung cancer in current and former smokers at high risk for this disease. The estimated date of completion is September 2021, with no clinical results available yet. 

Another important TAA is HER2, one of the best-known breast cancer-associated antigens. HER2 is already overexpressed in breast ductal carcinoma in situ (DCIS), the main precursor of invasive breast cancer (IBC). Moreover, HER2 is also overexpressed in other epithelial tumors such as gastric, biliary tract, colorectal, non-small-cell lung, and bladder cancers [47]. Trastuzumab, the first HER2-targeted agent approved for clinical use in breast cancer patients, demonstrated a significant overall survival improvement in HER2-positive early-stage and metastatic cancer patients in the early 2000s [48]. This success was followed by several drugs targeting the HER2 pathway. Significantly, it has been recently demonstrated that a loss of anti-HER-2 CD4+ Th1 responses in peripheral blood occurs during breast tumorigenesis which can be restored with HER-2 vaccinations in both DCIS and IBC [49]. Following this relevant finding, several trials focused on HER2-based vaccines in early pre-malignant lesions and in the prevention of recurrence of IBC patients as preventive approaches. Amongst them, a phase I/II clinical trial with a peptidic vaccine composed of E75 (nelipepimut-S, an immunogenic peptide from the HER2 protein) plus GM-CSF for the prevention of disease recurrence in high-risk breast cancer patients demonstrated safety and clinical efficacy in a 5 year follow up [50]. However, a phase III trial (2019) showed no significant difference in disease-free survival between vaccinated individuals and the placebo arm, and a high number of recurrences in the vaccine arm compared to placebo (54.1% in the vaccine arm vs. 29.2% in the placebo arm, *p* = 0.069) drove the trial to an early termination due to futility [51]. The authors specify that the use of mandated annual scans and image-only recurrence events hastened the interim analysis that drove the trial termination, however, the clinical significance of these radiographic-only findings was unclear, as patients were not followed after trial termination [51]. A more recent phase I trial started in 2016 (NCT02780401) is studying the side effects and best dose of a DNA plasmid-based vaccine encoding three breast cancer antigens (insulin-like growth factor-binding protein [IGFBP]-2, HER2, and insulin-like growth factor [IGF]-1 receptor [1R]) for the prevention of cancer recurrence in patients with a history of non-metastatic, node-positive, HER2 negative breast cancer. Also importantly, vaccines based on the AE37 peptide, which is derived from the intracellular HER2 portion fused with the MHC-II invariant chain li-key, have demonstrated safety and effectiveness in decreasing disease recurrence in selected breast cancer patients. As a promising example, in a phase II trial for an AE37 vaccine (NCT00524277), patients with advanced-stage breast cancer and HER2 low expression had a 5-year estimated DFS of 83% on the treated arm compared with a 62.5% in the GM-CSF alone arm (*p* = 0.039, HR 0.375, CI 0.142–0.988) [52,53,54]. 

### 3.2. Cancer-Testis Antigens (CTAs)

The melanoma-antigen family A proteins (MAGE-A), the Preferentially expressed Antigen in Melanoma (PRAME), and the New York esophageal squamous cell carcinoma 1 (NY-ESO-1) are amongst the principal targeted CTAs for cancer vaccines. PRAME is found to be overexpressed in melanomas, acute leukemia cells, various sarcomas [55], breast cancer, cervical cancer, lung cancer, melanomas, and ovarian cancers among others [56]. The PRAME immunotherapeutic (recombinant PRAME protein (recPRAME) with the AS15 immunostimulant) has yielded promising specific antibodies and CD4+ immune responses in patients with NSCLC [57] and with metastatic melanoma [58].

NY-ESO-1, a molecule with restricted expression pattern to germ cells and placental cells and re-expression in numerous cancer types, has been widely applied in various immunotherapeutic strategies due to its proven ability to elicit spontaneous humoral and cellular immune responses [59]. Many clinical trials have demonstrated that NY-ESO-1 vaccination induces specific CD4+ and CD8+ responses, and importantly, the vaccine with the 20-mer NY-ESO-191–110 peptide [60] and the dendritic cell-based therapy combining a NY-ESO-1 protein vaccine with liposome-encapsulated doxorubicin and decitabine [61] both induced stable disease at different degrees in patients with tumors that expressed the molecule (Esophageal, lung, gastric and ovarian cancers) [62]. Several reviews addressing results from clinical trials with this CTA state that the main challenge is to counter potential roadblocks in therapeutic pathways which mainly include (A) tumor immune evasion, (B) inefficient induction of high-affinity adaptive immunity, and (C) tumor immunosuppression. Given the positive clinical trial results and taking advantage of the restricted expression pattern of the molecule, these vaccines could be applied to individuals at high risk of cancers that typically express NY-ESO-1, thus avoiding most of the associated therapeutic limitations.

Although with less relevant clinical efficacy, there are also some promising trials featuring the MAGE-A proteins. Inside this family, MAGE-A1-4 proteins are CTAs with restricted expression in healthy tissues and wide expression in several carcinomas: a systematic immunohistochemical analysis of 3668 common epithelial carcinomas (CA) and germ cell tumors of high prevalence and mortality reported that MAGE-A is more highly expressed than NY-ESO-1 in tumors including esophageal squamous cell carcinomas (SCC), bladder urothelial CA, head and neck/cervix/anal SCC, lung SCC and adenocarcinomas, lung small cell carcinoma, ovarian CA, endometrial CA, hepatocellular CA, gastric adenocarcinomas, colorectal adenocarcinomas, and breast ductal carcinomas [63]. Despite the null clinical efficacy of a vaccine encompassing recombinant MAGE-A3 together with an immunostimulant (AS15), which was tested in two comprehensive randomized, double-blinded, placebo-controlled phase III clinical trials in patients with completely resected stage IB, II, and IIIA MAGE-A3-positive NSCLC (MAGRIT) [64] and in patients with completely resected, stage IIIB or IIIC, MAGE-A3-positive cutaneous melanoma (DERMA) [65], other formulations of this CTA are proving to be more effective. For instance, a multi-institutional phase 1/2 study in children with relapsed or refractory neuroblastoma or sarcoma demonstrated that their peptide-pulsed DC-based vaccine targeting MAGE-A1, MAGE-A3, and NY-ESO-1 in combination with decitabine elicited objective clinical responses: one patient had a complete response and another one had a partial response; moreover, six out of nine patients experienced specific T-cell responses, but notably none of them developed antibodies against the targeted antigens [66]. However, five patients suffered grade ≥3 adverse events, which denotes that clinical benefit vs toxicity needs to be further studied. A more recent in vitro study demonstrated that in samples from patients with esophageal SCC dual PD-1/PD-L1 and TGF-β signaling pathway blockades synergistically restored antitumoral properties of MAGE-A3-specific CD8+ T cells, suggesting that a combinatorial approach would be beneficial for these vaccines [67]. This would also imply that in a preventive approach where the immune-blockade pathways (such as PD1-PDL1, CTLA4, or TGFβ) are marginal, MAGE-A-based vaccines could be more efficient.

In conclusion, results from preliminary TAA and CTA trials demonstrate the potential immunological benefit of vaccines targeting these molecules in secondary preventive settings, where boosting immune responses against them can generate long-lasting memory that may decrease disease recurrence. The role of these approaches in primary prevention in individuals with pre-malignant lesions, where pre-existing cellular and humoral immune responses are taking place due to recognition by immune surveillance [68], remains still largely unexplored. Furthermore, evidence of immunosuppression and immune escape in both the preventive and therapeutic setting suggest that combinatory vaccination with checkpoint inhibitors could be an optimal immunopreventive strategy, similar to what has been demonstrated in immunotherapy settings [69].

The combination of better screening procedures, in combination with preventive immunological vaccines in healthy individuals at high risk of tumors expressing determinate TAAs would be transformative in cancer prevention.

### 3.3. Tumor Specific Antigens and Neoantigens

With the advent of next-generation sequencing (NGS) and the overall improvement in molecular biology, interest in personalized medicine is increasing exponentially. The workflow for personalized neoantigens-based vaccines strategies roughly includes obtention of pre-neoplastic or tumoral samples, DNA and/or RNA extraction, and sequencing followed by an in silico determination (or prediction) of neoantigens expressed in that particular patient, taking into account their HLA haplotype. This allows the generation of an individualized vaccine targeting highly specific tumor-associated neoantigens or neoepitopes [23].

Healthy individuals with an intact immune system and a high risk of cancer are the ideal targets of prophylactic vaccination in terms of eliciting optimal vaccine potential. As in these cases, personalized medicine (personalized neoantigens vaccines, Table 1) is difficult to implement because of the lack of pre-neoplastic or neoplastic lesions from which to predict the optimal vaccine targets, a practical approach would be an off-the-shelf vaccine against predictable mutations in putative oncogenes to yield broad-spectrum protection (shared or common neoantigens-based vaccine, Table 1). The poor overall survival benefits in neoantigen-based therapeutic vaccines are generally attributed to immunologic exhaustion, the immunosuppressive tumoral microenvironment, immunoediting events, and tumor burden [70]. Although several clinical trials have shown that combinatorial therapies [71,72] could partially overcome these challenges, the full potential of neoantigen-based vaccines would be in a preventive approach.

Several studies have shown that from the multiple antigens derived from a determinate gene or tumor sample predicted in silico to be immunogenic, only a few elicit optimal immunologic responses in in vitro functional screenings [26,73]. Thus, tumors with a low mutational load have a low probability to generate and present cancer-specific neoantigens. Indeed, it has been demonstrated that patients with lung cancers or melanomas with a high mutational load experience a higher rate of response to immune checkpoint blockade [7], and in pancreatic cancer, it was recently determined that tumors with both the highest neoantigen number and the most abundant DCs and CD8+ T-cell infiltrates corresponded with the longest patient survival [74]. In this regard, cancers with microsatellite instability (MSI) are a more adequate target for CPVs as they accumulate more than 10-fold higher numbers of somatic mutations than microsatellite-stable (MSS) tumors [75].

Microsatellites are short tandem DNA repeats of 1 to 6 base-pair motifs that are dispersed in the eukaryotic (and in less proportion in the prokaryotic) genome, accounting for up to 1–3% of the human genome [76]. Due to their repetitive structure, in these sequences, the DNA polymerase has a higher proportion of slippages during replication, namely single nucleotide variants or insertions and deletions (Indels) of 1 to several bases. The mismatch repair (MMR) system, formed by MLH1, MSH2, MSH6, and PMS2, is a cellular homeostasis-preserving post-replication process that, if deficient (MMRd), fails to correct the previously mentioned errors leading to a phenotype of genome instability known as MSI-H (high) [75]. Even though microsatellites are more frequent in intergenic and non-coding regions, there are a few genes that have them in their coding sequence where the consequence of a non-synonymous mutation is a shift on the open reading frame (ORF) that generates a highly immunogenic c-terminus mutated protein (frameshift peptide, FSP, or neopeptide) with no previous central immune tolerance [77,78]. According to a recent study, FSP are three times less similar to viral epitopes than missense-derived neoantigens probably due to host–virus co-evolution and viral mimicry of host function, which would mean that FSP are even ‘‘further from self’’ than viral antigens [73].

MMRd tumors are characterized by a high mutational burden, increased lymphocytic infiltration, and the presence of prostaglandins and inflammatory cytokines. As mentioned previously, the increased inflammatory microenvironment and infiltration in comparison to MSS cancers have long been associated to the highly immunogenic neoantigens arising from MSI-H mutations and to a better prognosis in many cancer types [29,79]. Moreover, it is because of the increased presence of infiltrating cancer-specific effector T-lymphocytes that MSI tumors respond better to checkpoint-inhibitor-based immunotherapies [73,80,81]. Due to the limited number of genes with microsatellite in their ORF, neopeptides derived from this path are thought to have a high probability of being common among multiple patients, which opens the possibilities to develop prophylactic or therapeutic off-the-shelf vaccines against them. In fact, a recent study by Roudko et al. [73] has identified tumor-specific frameshifts encoding multiple epitopes originated from InDel mutations shared among patients with MSI-H endometrial, colorectal, and stomach cancers, suitable for the design of common ‘‘off-the-shelf’’ cancer vaccines. On an important note, and in contradiction with the previous observations, the selected full FSPs in this study primarily elicited CD8+ cytotoxic responses in both healthy donors and MSI-H patients [73] instead of CD4+, which could be attributed to their prediction and selection strategy of FSP that specifically bind multiple MHC-I molecules.

The fact that neoantigens appear early in preneoplastic lesions although in lesser numbers than in tumors [75,82] strengthens the feasibility of a prophylactic vaccination for the prevention of pre-neoplastic lesions that derive into tumors.

MSI can occur spontaneously in 10–15% of colorectal and up to 25 to 35% of endometrial cancers, mainly due to somatic inactivation of the MMR genes by either biallelic hypermethylation of the MLH-1 promoter or by double somatic mutations in one of the MMR system genes. Remarkably, tumors in the context of Lynch syndrome (LS, formerly known as hereditary nonpolyposis colorectal cancer, HNPCC), the most common inherited cancer syndrome, are characterized by MMRd [83,84,85]. LS is caused by monoallelic germline mutations in one of the MMR genes (MLH1, MSH2, MSH6, and PMS2), which predisposes this population to a high risk of developing MSI-H cancers—mainly CRC (accumulated risk of 40–80%) and endometrial cancer (40–60%), usually at young ages [86]. Although the tumor spectrum is broader (stomach, urinary tract, ovary, small intestine, skin, CNS), these neoplasms are much less frequent (cumulative risk < 10%) [78]. Because of its defined associated status of high risk, LS is a unique scenario to study and establish preventive cancer vaccines. The prevention strategy in LS CRC consists of screening colonoscopies every 1–2 years from 20–25 years, to detect pre-malignant lesions or cancer in an early stage [87]. This strategy has been shown to significantly decrease incidence and mortality by CRC. However, despite colonoscopies, up to 45% of patients will develop a second tumor in the colon, which sometimes requires a second surgery. In relation to endometrial and ovary tumors, preventive hysterectomy and double anexectomy is the only strategy that has been demonstrated to reduce the risk of endometrial cancer [88]. More importantly, there are no proven prevention measures for the other tumors of the syndrome.

Many recent studies have worked to identify shared neoepitopes in this context and have started to develop immunotherapies targeting them: Kloor et al. [26] recently shared their phase I/IIa clinical trial results on the vaccine Micoryx. The vaccine is based on three FSP derived from mutant *AIM2*, *TAF1B*, and *HT001* genes, which are commonly mutated among tumors with MMRd. Vaccination in 22 patients with a history of MMRd CRC indicated safety and tolerability together with specific humoral and cellular immune responses, although clinical efficacy of the vaccine could not be examined due to 19 patients being tumor-free at study inclusion. However, they remarked the high potential of the FSP vaccine for tumor prevention in Lynch syndrome. Remarkably, this group recently published a complementary study in mice models where they tested an FSP vaccine consisting of four shared FSP neoantigens derived from mouse coding mononucleotide repeats, which were in silico predicted and in vivo validated in naïve C57BL/6 mice. The vaccine effectively reduced intestinal tumor burden and prolonged overall survival in VCMsh2 mice, which have a conditional knockout of Msh2 in the intestinal tract and develop intestinal cancer [89]. Their findings prove that a recurrent FSP neoantigen vaccination for Lynch Syndrome cancer immunoprevention needs to be further evaluated.

The beneficial effect of neoantigen-based approaches is also well established in melanoma: a recent trial has evaluated the clinical outcome and circulating immune responses of eight patients with surgically resected stage IIIB/C or IVM1a/b melanoma, at a median of almost 4 years after treatment with NeoVax, a long-peptide vaccine targeting up to 20 personalized neoantigens per patient and formulated with the TLR-3 and MDA5 agonist poly-ICLC (NCT01970358) [22]. TLR3 is a receptor found on endosomal compartments that detects double-stranded RNA, which is produced in the majority of viral infections [90]. Poly(I:C) binds to TLR3 and induces the activation of the NF-κB and mitogen-activated protein (MAP) kinases independently of MyDd88 and TRIF-dependent, causing dendritic cells to mature and release pro-inflammatory cytokines [91].

Previous evidence showed that the NeoVax vaccine is feasible, safe, and immunogenic in patients with high-risk melanoma [92], and in patients with glioblastoma [93]. In the latest trial, they show NeoVax-induced memory T cell responses that exhibit cytolytic properties in vivo and are sustained in peripheral blood over a median of four years. Additionally, they have evidence of de novo T cell responses directed against neoantigen targets and TAAs that were expressed by the tumors but not contained in the vaccines. Regarding the main clinical outcome consisting of the number of patients alive and with no progression after surgery and vaccination, they report that after a four-year follow-up all patients were alive and six were without evidence of active disease [22]. Another recent report reviewed the 12-year survival of patients with non-resectable metastatic melanoma who received a vaccine composed of monocyte-derived DCs loaded with four MHC class I-binding and six MHC class II-binding melanoma-derived peptides, which was injected intradermally over a period of 2 years [94]. They report that up to 19% of these patients were still alive, an outcome that is comparable to the results of checkpoint blockade treatment in melanoma (CTLA4) [95].

Although the targeting of neoantigens in CPVs is attractive because of biological and practical reasons, namely the high immunogenic potential and the prevalence among individuals which makes it possible to produce off-the-shelf vaccines, there are several limitations in their development. The first one is the restricted subset of tumors where these vaccines could be applied, which limits to cancers with a high mutational load such as melanoma, NSCLC, MSI-h CRC, bladder, and stomach cancers, among others. Secondly, due to the high diversity of HLA molecules many trials choose to reduce their target population to that with a certain HLA allele in order to make neoantigen predictions more feasible and specific, thus prioritizing immunogenicity over a range of protection. In order to cover a wide range of HLA alleles requires the use of several neoantigens [22]. However, vaccine efficacy could be enhanced depending on the carriers and/or adjuvants during manufacturing; for example, in vaccines based on dendritic cells, these can be separately saturated with a certain peptide to avoid competition amongst different HLA binding affinities, ensuring that all epitopes included in the vaccine will be presented [96].

## 4. Challenges and Approaches for the Identification of Neoantigens

A crucial point when designing a neoantigen-based vaccine is the type of prediction applied for the selection of candidates. Usually, the strategy for in silico prediction of neoantigen candidates takes into account several biological key points, from the identification of various somatic mutations to MHC-TCR interactions (summarized in Figure 2):

Matched tumor–normal DNA sequencing data from whole-exome or genome sequencing (WES, WGS) are analyzed to call somatic mutations through variant caller software. In parallel, the patient’s HLA alleles should be determined by HLA haplotyping. Additional RNA sequencing (RNA-seq) may serve to quantify and verify gene and transcript expression, as it can identify gene fusions, alternative splicing isoforms, and other RNA editing events that WES cannot. After mutation analysis, candidate neoantigens can be identified through a variety of pipelines that include binding prediction algorithms to MHC class I and II molecules as well as additional and optional predictors of overall patterns of gene expression, RNA-splicing, peptide processing steps such as proteasome proteolysis, efficiency and affinity of peptide loading into TAP proteins, which are the intermediary present on the endoplasmic reticulum previous to MHC-I loading, and predictors of secondary structure degradation for MHC-II peptides (Figure 1) [97]. Also importantly, the inclusion of prediction of mRNA decay escape has recently been shown to associate with higher antitumor immunogenicity [98].

Many studies focus on MHC-I restricted predictions because of the limited accuracy of current MHC-II prediction algorithms. Although MHC-I molecules and associated peptide processing have been extensively characterized, only recently the focus has shifted to class-II-specific neoantigens because of promising results in cancer immunotherapies and the determination of the importance of CD4+ T-cells activation for the achievement of a complete antitumoral response [31]. Additionally, the lack of reliability of MHC-II binding predictors is due to the promiscuous nature of the molecule which can hold different protein lengths (13 to 25 residues) in its multivariable open-ended groove. MHC-I molecules are more restrictive, and it is established that presented peptides range from 8 to 11 amino acids [99].

The use of mass spectrometry (MS) has improved prediction algorithms by providing information on peptides bound to MHC proteins, and consequently a deeper understanding of endogenous peptide processing and presenting. This, although still restricted to certain alleles, has considerably enhanced MHC binding prediction reliability. However, further studies are needed to increase the number of alleles on which these algorithms are trained and to better understand the factors involved in antigen expression, presentation, and immunogenicity [23].

Importantly, in silico predictions, target discovery, and prioritization are crucial to determine a candidate set; however, it does not provide direct information on whether these antigens will be able to induce T-cell responses. Typically, T-cell priming results in TNF-a and/or IFN-ƴ cytokine responses, IL-2 release, T cell proliferation, and the acquisition of cytolytic activity in the case of CD8+ T cells [29]. It is necessary to test these aspects with in vitro functional assays through different experimental techniques, including (A) immunogenic cytokine production assessment upon exposure to antigens by ELISA, ELISPOTs, Luminex technology, or intracellular staining; (B) binding to synthetic tetrameric or dextrameric complexes of peptide-MHC (pMHC) molecules; and (C) T-cell and APCs polarization and maturation status through flow cytometry, immunohistochemistry, and RNA expression, among others [100].

In prophylactic vaccines it is also important to validate antigens in cells derived from both healthy individuals and patients with ongoing or with a history of the studied tumor, also considering HLA haplotypes of the samples used for screening in order to avoid related biases in the results. Antigen-specific T-cells have been identified within peripheral blood mononuclear cells (PBMC) and tumor-infiltrating lymphocytes, which given their presence in the tumor and presumed clonal expansion by antigens presented in the tumor microenvironment, are the preferred method for recognition studies [101,102]. Studies of in vitro priming on cells derived from healthy donors are key in preventive vaccines as they give information on whether a determinate antigen is capable of being immunogenic without previous contact. An interesting novel method for these studies is the Genocea’s ATLASTM bioassay, where autologous APCs expressing every predicted peptide (using Escherichia coli) are incubated with autologous PBMC-derived T-cells to assess cytokine production and validate their immunogenicity [103].

## 5. Vaccine Vectors

Choosing the right vector or delivery system for a vaccine is determinant for the induced immunological responses. Delivery platforms for CPVs currently include synthetic peptides, viral vectors, cellular vaccines, and nucleic acids (DNA and mRNA) [12,104], advantages and disadvantages for each one are summarized in Figure 3. In therapeutic cancer vaccines, a key factor for the vector platform choice is the time needed to design, manufacture, and administrate the vaccine, as there is an urgency related to the patient’s disease. However, this issue is secondary in CPVs. Another advantage of prophylactic vaccination is the accessibility of target immune cells due to the absence of tumor burden, which makes the biological being tested more crucial than delivery systems when trying to induce an optimal antitumoral response.

### 5.1. Nucleic Acid Vaccines and Peptides

In peptidic, mRNA and DNA vaccines, in general, administration of soluble or “naked” antigens or molecular adjuvants show very poor targeting efficiency and limited immune responses due to rapid dissemination into the systemic circulation [104,109]. Moreover, mRNA can be spontaneously taken up by a variety of cells through endocytosis, leaving only a small part for the capturing by APCs and its subsequent translation and antigen processing. Strategies to protect mRNA and DNA molecules such as encapsulation within lipidic or polyethylene glycol-coated nanospheres improve the stability and efficacy of these molecules. Vaccine efficiency can also be dramatically improved by the combination with immunogenic adjuvants, such as the successful TLR3 agonist poly ICLC [46,92,110] (NCT02950766) or the granulocyte-monocyte colony stimulation factor (GM-CSF). In a recent trial in mouse models, prophylactic vaccination of exosomes derived from murine embryonic stem cells (ESCs) engineered to produce GM-CSF significantly blocked outgrowth of implanted lung carcinoma and induced robust CD8+ and Th1 responses [111]. Other constructs include amphiphilic peptides, combinations with other immunomodulators [112] as well as different carriers to enhance uptake by APCs especially on mRNA and DNA vaccines. In these lines, several delivery systems have been tested and demonstrated to improve transfection and thus leading to stronger immune responses, including nanoparticles, liposomes, microneedle arrays, gene gun, and in situ electroporation [113]. In a recent phase I trial in advanced melanoma patients, vaccination with liposomes loaded with mRNA expressing four different TAAs induced specific T-cell responses and either disease stabilization or regression of metastatic lesions [114].

#### Virus-like Particles

Viral vectors or virus-like particles-based vaccines have also been deeply studied as vector platforms because of their inherently immunogenic nature and their ability to easily introduce genetic material into cells [115]. In a recent pre-clinical trial, a novel replication-defective recombinant adenovirus 40 (rAd40) expressing mouse mesothelin (Msln) was administrated intravenously in mouse models as a prophylactic cancer vaccine against metastatic lesions of pancreatic cancer. Msln protein expression and metastases were suppressed in a syngeneic orthotopic mouse model of pancreatic cancer, corresponding to the detection of Msln- and tumor-specific CD8+ T-cells [116]. One of the disadvantages of this strategy is the neutralization of viral vectors by the immune system. To solve this problem, some studies have proposed a prime-boosting strategy consisting of two phases: firstly, priming with either in situ delivery of tumor antigens (i.e., tumor lysates or synthetic antigens) or ex vivo DCs priming; and secondly, boosting by the administration of a personalized synthetic vaccine consisting of those antigens that had generated the most potent immune responses [117]. In a phase II randomized trial for the treatment of Metastatic Castration-Resistant Prostate Cancer (mCRPC), the PROSTVAC (prostate-specific antigen [PSA]-TRICOM) vaccine was administered with a regimen of prime-boosting where a vaccinia virus encoding the PSA antigen was used for priming followed by six subsequent booster doses of a fowlpox virus encoding PSA [118]. Unfortunately, in a phase III trial the same strategy failed to reproduce the findings of OS improvement that the phase II trial had demonstrated. However, thanks to the positive results from phase II and data suggesting that PSA-TRICOM vaccine may have greater efficacy in earlier-stage diseases [119], there have been several studies that continue testing this strategy in different settings: In a randomized phase II trial (NCT02326805), the National Cancer Institute is studying how well the PROSTVAC (prostate-specific antigen [PSA]-TRICOM) vaccine works in preventing disease progression in patients with prostate cancer undergoing active surveillance. In this case, the vaccine consists of a pox-viral vector encapsulating the PSA and three T-cell costimulatory molecules (B7.1, ICAM-1, and LFA-3). Additional studies with checkpoint inhibitors are also being conducted (NCT02933255).

### 5.2. Cellular Vaccines

Cellular-based vaccines comprise both the use of whole-tumor-cell lysates or antigen-loaded autologous antigen-presenting cells (or a combination of both), most commonly dendritic cells. In the case of prophylactic settings, except in pre-malignant lesions, secondary and tertiary prevention levels, a tumor cell lysate cannot be obtained; thus, the use of autologous antigen-loaded cells is far more feasible.

DCs have been extensively applied in cancer immunotherapy, with more than 200 trials completed (reviewed in [32]). These approaches generally involve the ex vivo generation or isolation of autologous (circulating or monocyte-derived) DCs from an individual’s cytapheresis and their subsequential maturation, antigen-priming (with mRNA, DNA, peptides, tumoral cell lysates, etc.), and reinfusion to the patient [96]. The extensive experience with this vaccination strategy has demonstrated that it is clinically safe and well tolerated with minimal side effects (mostly reported grade I and II adverse events, with only up to 3–4% reported grade III adverse events), highly immunogenic in terms of induction of specific CD4+ and CD8+ responses and, importantly, capable of inducing notable response rates in therapeutic settings; however, clinical effectivity remains low. This can be, among other factors, due to the patient’s immune system exhaustion at the time of treatment and the transition of DCs from an in vitro to an in vivo immunosuppressive environment which can affect viability and functionality [32,104,120]. The full potential of DC-based vaccines would most probably be unleashed in a preventive setting.

Regarding preventive vaccination, a recent phase I/II open-label clinical trial from Radboud University (NCT01885702) studied the safety, feasibility, and induction of specific immune responses (as well as the pathological and clinical responses) of a vaccine based on frameshift-derived neoantigen-loaded DCs from CRC patients with an MSI-positive CRC or Lynch syndrome mutation carriers. Although no definitive clinical results are available yet, they have reported that the vaccine is safe with minimal elicited side effects, and that specific CD4+ and CD8+ immune responses were induced against the target peptides (TGFβRII (RLSSCVPVA) and caspase-5 (FLIIWQNTM)). These results represent the proof-of-concept that neoantigen-loaded DC-based preventive antitumoral vaccination is safe and feasible.

## 6. Concluding Remarks

Despite the current therapeutic effort, cancer is still the second leading cause of death worldwide. It is clear that the tumor microenvironment and the considerable heterogeneity of tumor cells, which have just begun to be addressed in recent years, substantially hamper the effectivity of even the most advanced treatments. Cancer prevention through vaccination is a novel approach to tackle cancer that would leverage the high antitumoral potential of current cancer vaccines without being affected by tumor burden. Over the last years, deeper insights into the rising field of immuno-oncology have prompted researchers to start focusing on CPVs applied to individuals with high genetic cancer risk or suffering from pre-malignant lesions to generate de novo responses or to strengthen immune surveillance, and several trials are underway that are yielding promising results. The years to come will show if these new approaches can hold promise and whether CPVs could be routinely applied to high-risk individuals in a not too far future. In these lines, we propose that individuals with Lynch Syndrome are an excellent clinical model to study preventive approaches.

## Figures and Tables

**Figure 1 ijms-22-10900-f001:**
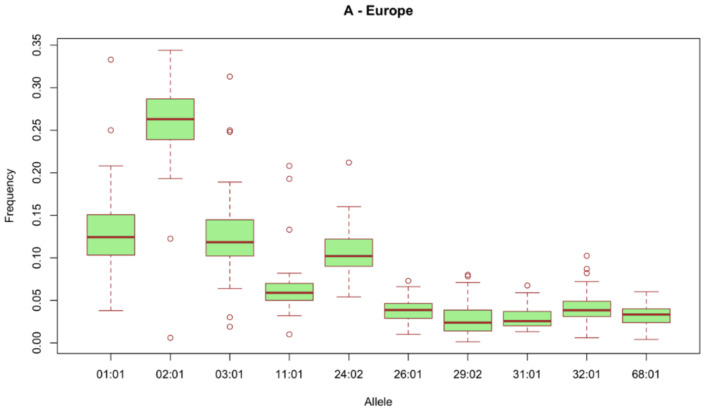
Overall estimation of frequencies for most common HLA-A alleles by geographical region (Europe). Taken from Allele frequency net database (AFND) 2020 [28].

**Figure 2 ijms-22-10900-f002:**
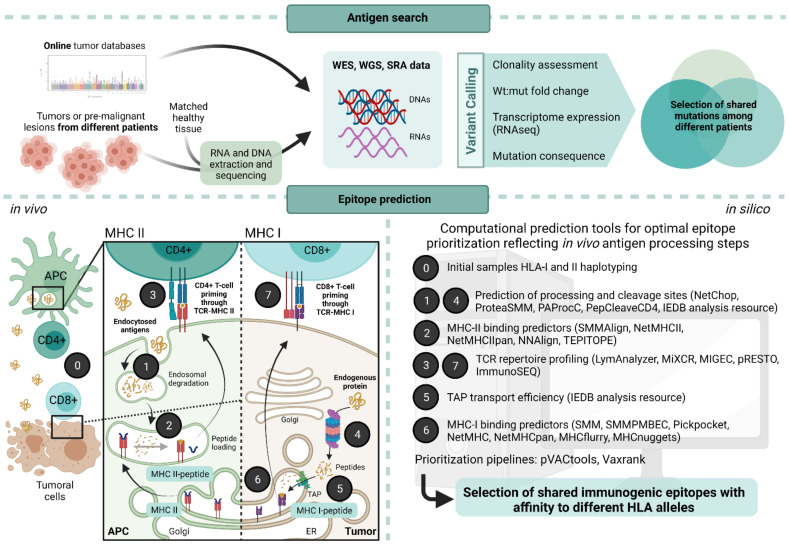
Schematic representation of antigen prediction pipeline for the design of cancer preventive vaccines (CPVs). **Above**: Workflow to determine shared mutations among different tumors and patients for a preventive and off-the-shelf approach. **Below**: Biological key steps in antigen presentation pathways (MHC-I and MHC-II) to consider when performing in silico epitope predictions. Abbreviations: WES, whole exome sequencing; WGS, whole genome sequencing, SRA, short read alignment; Wt, wildtype; APC, antigen-presenting cell; ER, endoplasmic reticulum; TAP, transporter associated with antigen processing. Created with BioRender.com.

**Figure 3 ijms-22-10900-f003:**
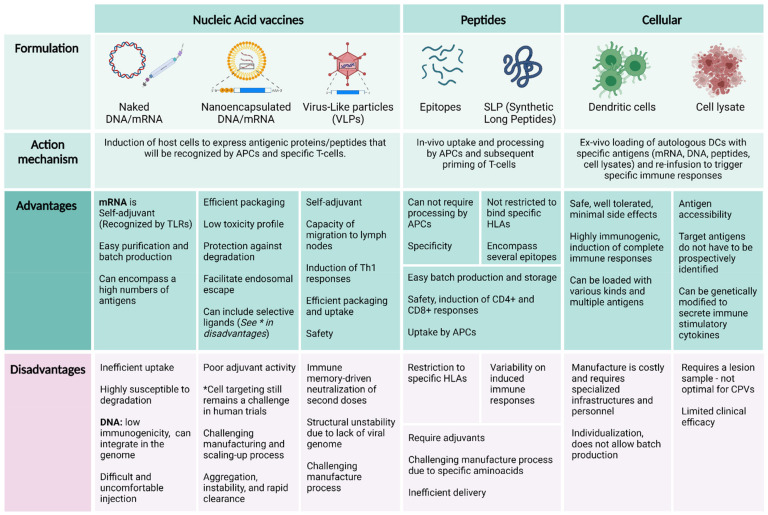
Advantages and disadvantages of cancer prevention vaccine vectors. Abbreviations: mRNA, messenger RNA; TLRs, toll-like receptors; APC, antigen-presenting cell; HLA, human leukocyte antigen; CPV, cancer prevention vaccine. References [11,12,44,105,106,107,108]. Created with BioRender.com.

**Table 1 ijms-22-10900-t001:** Antigen targets for cancer prevention vaccines.

Target Antigen	Examples	Tumor Specificity	Central Tolerance	Prevalence among Patients	Optimal Preventive Setting	Suitable for CPV
Tumor associated antigens (TAAs)	MUC1 (epithelial tumors: colon, pancreatic, cervix), HER2 (breast, bladder, gastric cancers), EGFR (NSCLC, glioma), CEA (colorectal, pancreatic, gastric, lung and breast cancers), cyclin B1 (gynecological and colorectal cancers)	Medium	Yes	Variable	Individuals with genetic predisposition, pre-malignant lesions, prevention of recurrences	Yes
Cancer testis antigens (CTAs)	NY-ESO (melanoma and carcinomas of lung, esophageal, liver, gastric, prostrate, ovarian, and bladder), MAGE-A 1–4 (NSCLC, bladder, esophageal and head and neck cancers, sarcomas, triple negative breast cancers, myeloma, Hodgkin’s disease), PRAME	Medium	Partial	High	Individuals with genetic predisposition or risk practices (such as smoking), pre-malignant lesions, prevention of recurrences	Yes
Shared or common neoantigens	Mutated oncogenes, passenger/driver mutations in Lynch syndrome (AIM2, ACVR2A, TGFBRII, CASP5, …)	High	No	High	Individuals with genetic predisposition (such as Lynch syndrome), pre-malignant lesions	Yes
Personalized neoantigens	Depending on each tumor and patient	High	No	Very low	None	No

CEA, carcinoembryonic antigen; EGFR, epidermal growth factor receptor; HER2, human epidermal growth factor receptor 2; MAGEA, melanoma-associated antigen; MART1, melanoma antigen recognized by T cells 1; MUC1, mucin 1; NSCLC, non-small cell lung cancer; PRAME, melanoma antigen preferentially expressed in tumors.

## Data Availability

Not applicable.

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
