# Peer review of "Vaccines for Non-Viral Cancer Prevention"

_ijms, 2021, doi:10.3390/ijms222010900_

Round 1

Reviewer 1 Report

In this review manuscript, the authors provided the overview of cancer prophylactic vaccine. The review article includes the principle of cancer vaccine, types of antigens, and vectors for vaccine delivery. Recent advance technologies in preventive cancer vaccines are also described in this manuscript. This will be of interest to many readers not only in cancer vaccinology area, but for a general prophylactic vaccinology.

There are some points of concerns in this manuscript.

  1. Under section 3, Antigen Targets. The authors classified target antigens into four types (Table.1). However, the authors combined CTAs to TAAs (3.1) and combined shared or common neo-antigens to personalized neo-antigens. It should be better if the authors could provide segments in section 3 by following the classification in Table.1. Moreover, no description of MEGEA 1-4 and PRAME antigens in section 3. 
  2. There are errors in numbering under section 5. In section 5, vaccine vectors, the numbers becomes 4.1 (line 565, line 601, line 625).
  3. The authors used Figure 3 to represent types of vaccine vectors. The table begins with nucleic acid and viral vectors, peptides and cellular vector respectively. However, in the text the authors categorized the vectors differently from Figure 3. It would be better if the authors categorize vectors in the text in agreement with Figure 3. 

Author Response

Dear reviewer,

We are very thankful for your commentaries about our review. We have assessed the mentioned points of concern and made changes to solve them:

  1. In section 3, nomenclature for the classification for target antigens has been unified in the text and the table 1. A paragraph addressing MAGE-A and PRAME has been added.
  2. Numbering errors in section 5 have been solved.
  3. Nomenclature for vaccine vectors has been unified between figure 3 and the text. Moreover, sections have been re-organized to follow the order established in figure 3.

Reviewer 2 Report

This is a extermely well-written review paper focused on vaccine approach for cacner prevention. This review addresses important aspects in an era of immunotherapy. It reads very well, and the overall structure/Figures are well-organized.   I strongly recommned this for publication without change.

Author Response

Dear reviewer,

Thank you so much for your commentaries about our review. We have implemented a few changes which we summarize here: 

  1. In section 3, nomenclature for the classification for target antigens has been unified in the text and the table 1. A paragraph addressing MAGE-A and PRAME has been added.
  2. Numbering errors in section 5 have been solved.
  3. Nomenclature for vaccine vectors has been unified between figure 3 and the text. Moreover, sections have been re-organized to follow the order established in figure 3.

We are very thankful and look forward to publishing our review.